# Relationship between Vitamin Intake and Health-Related Quality of Life in a Japanese Population: A Cross-Sectional Analysis of the Shika Study

**DOI:** 10.3390/nu13031023

**Published:** 2021-03-22

**Authors:** Nobuhiko Narukawa, Hiromasa Tsujiguchi, Akinori Hara, Sakae Miyagi, Takayuki Kannon, Keita Suzuki, Yukari Shimizu, Thao Thi Thu Nguyen, Kim Oanh Pham, Fumihiko Suzuki, Atsushi Asai, Takashi Amatsu, Tomoko Kasahara, Masateru Miyagi, Masaharu Nakamura, Yohei Yamada, Haruki Nakamura, Koichiro Hayashi, Toshio Hamagishi, Aki Shibata, Tadashi Konoshita, Yasuhiro Kambayashi, Hirohito Tsuboi, Atsushi Tajima, Hiroyuki Nakamura

**Affiliations:** 1Department of Environmental and Preventive Medicine, Graduate School of Medical Science, Kanazawa University, 13-1 Takaramachi, Kanazawa, Ishikawa 920-8640, Japan; t-hiromasa@med.kanazawa-u.ac.jp (H.T.); ahara@m-kanazawa.jp (A.H.); kimoanhpham129@gmail.com (K.O.P.); f-suzuki@den.ohu-u.ac.jp (F.S.); vetasai1998@gmail.com (A.A.); takashi.kamisama@icloud.com (T.A.); kenshoudou@mx3.et.tiki.ne.jp (T.K.); okitanpopo@yahoo.co.jp (M.M.); m.nakamura.83-7-7@r.vodafone.ne.jp (M.N.); yamada503597@stu.kanazawa-u.ac.jp (Y.Y.); haruki_nakamura_kanazawa@yahoo.co.jp (H.N.); orihciok1003@gmail.com (K.H.); hamagisi@chubu-gu.ac.jp (T.H.); akintoki1116@gmail.com (A.S.); hiro-n@po.incl.ne.jp (H.N.); 2Department of Public Health, Graduate School of Advanced Preventive Medical Sciences, Kanazawa University, 13-1 Takaramachi, Kanazawa, Ishikawa 920-8640, Japan; keitasuzuk@stu.kanazawa-u.ac.jp; 3Advanced Preventive Medical Sciences Research Center, Kanazawa University, 1-13 Takaramachi, Kanazawa, Ishikawa 920-8640, Japan; kannon@med.kanazawa-u.ac.jp (T.K.); atajima@med.kanazawa-u.ac.jp (A.T.); 4Innovative Clinical Research Center, Kanazawa University, 13-1 Takaramachi, Kanazawa, Ishikawa 920-8641, Japan; smiyagi@staff.kanazawa-u.ac.jp; 5Department of Bioinformatics and Genomics, Graduate School of Advanced Preventive Medical Sciences, Kanazawa University, 13-1 Takaramachi, Kanazawa, Ishikawa 920-8640, Japan; 6Faculty of Health Sciences, Department of Nursing, Komatsu University, 14-1 Mukaimotorimachi, Komatsu, Ishikawa 923-0961, Japan; h_zu@me.com; 7Faculty of Public Health, Haiphong University of Medicine and Pharmacy, Ngo Quyen, Hai Phong 180000, Vietnam; nttthao@hpmu.edu.vn; 8Division of Dental Anesthesiology, Department of Oral Surgery, Ohu University School of Dentistry, 1-31, Misumidou, Tomitamachi, Kohriyama, Fukushima 963-8611, Japan; 9Third Department of Internal Medicine, University of Fukui Faculty of Medical Sciences, 23-3 Matsuoka Shimoaizaki, Eiheiji-cho, Yoshida-gun, Tsuruga, Fukui 914-0055, Japan; konosita@u-fukui.ac.jp; 10Department of Public Health, Faculty of Veterinary Medicine, Okayama University of Science, 1-3 Ikoinooka, Imabari, Ehime 794-8555, Japan; y-kambayashi@vet.ous.ac.jp; 11Institute of Medical, Pharmaceutical and Health Sciences, Kanazawa University, 1 Kakuma-machi, Kanazawa 920-1192, Japan; tsuboih@p.kanazawa-u.ac.jp

**Keywords:** health-related quality of life, vitamin, Short Form-36

## Abstract

Although epidemiological studies revealed a relationship between psychosocial states, such as depressive symptoms, and nutritional intake, limited information is currently available on vitamin intake. The Short Form-36 Health Survey (SF-36) is not limited to a specific disease, it is constructed based on a universal concept of health and is used to evaluate the Quality of life (QOL). A three-component scoring method was developed for “Physical component score (PCS)”, “Mental component score (MCS)”, and “Role/social score (RCS)”. Collectively, these summary scores are called the “QOL summary score”, which is regarded as a more detailed health summary score. In the present study, we aimed at epidemiologically examine the relationship between vitamin intake and QOL in middle-aged and elderly population in 3162 residents in Japan. In women, a multiple regression analysis showed a positive correlation between all vitamin intake and PCS scores, and between vitamin B6, folic acid, vitamin C, and MCS scores. In consideration of depression as MCS of SF-36 and chronic pain as PCS, an insufficient vitamin intake may affect QOL in women; however, a causal relationship has not yet been demonstrated.

## 1. Introduction

The proportion of the global population older than 60 years is expected to double by 2050, from 1.4 billion in 2015 to 2.1 billion [1]. Due to aging of the global population, interest in the health of middle-aged and elderly individuals has increased. According to the WHO [2], “health is a state of physical, mental and social wellbeing, not simply the absence of disease”. In addition to disease onset and death, it is also important to evaluate life function and quality of life (QOL) from physical, mental, and social aspects. From this background, as in this study, are the three components of SF36, which are health-related QOL components score, (Physical Component summary; PCS), the mental QOL component score. (Mental Component summary; MCS), and the social-related QOL component score (Role Component summary; RCS), of the role aspect. Various environmental factors and lifestyles influence health- related QOL. Previous studies demonstrated that diet affected the QOL of middle-aged and elderly individuals. Proper eating habits have been identified as an important factor influencing physical and mental health, are closely related to QOL, and improve not only physical health, but also attenuate mental and emotional instabilities [3]. In addition, a relationship was reported between a regular diet and PCS and MCS in middle-aged and elderly individuals in China, indicating the importance of a proper diet for maintaining QOL [4]. However, limited information is currently available on the effects of dietary vitamins on QOL. Therefore, more detailed nutritional epidemiology is required to elucidate the relationship between a diet including vitamins and QOL in middle-aged and elderly individuals [5,6,7,8]. In the present study, we aimed at epidemiologically examine the relationship between vitamin intake and QOL in middle-aged and elderly population.

## 2. Materials and Methods

### 2.1. Participants

We utilized cross-sectional data from the Shika study [9,10,11]. Participants were recruited between October 2013 and December 2016. The target population was residents living in 4 model districts (Horimatsu, Higashimasuho, Tsuchida, and Togi districts) in Shika town, Ishikawa prefecture, Japan (population, 21,061, population aged 65 and older on 1 September 2020, 8499 (aging rate 42.2%)). The 4 districts were randomly selected, and about half of the population of the town live in the districts. A total of 5013 residents aged 40 years and older lived in the model districts. Written informed consent was obtained from 4724 participants (recovery rate 94.2%). Of these, 3202 participants answered SF-36 at a rate of 80% or higher with a reported energy intake of more than 600 kcal and less than 4000 kcal, with age range from 40 to 99. Therefore, 3202 participants (1429 men and 1773 women) were ultimately included in the analysis.

### 2.2. Evaluation of QOL

The Japanese version of SF-36 version 2 was used to measure the health-related QOL of participants. SF-36 is used specifically to assess QOL in a general population. It is one of the most widely and internationally used health-related QOL scales. The reliability and validity of the Japanese version have been examined [12]. This version has the advantage of being applicable to a healthy general population, regardless of the types of diseases among participants, unlike patient-specific scales, because it is a comprehensive QOL scale. It is composed of 8 subscales, with total scores ranging between 0 and 100. SF-36 measures eight health measures in the previous four weeks: Physical function (PF), role physical (RP), body pain (BP), general health (GH), vitality (VT), social functioning (SF), daily role (Mental) (role emotional; RE), and mental health (MH). Based on the Japanese national score, all scores were converted to relative scores (mean score; 50, standard deviation; 10), with higher scores indicating better QOL. These eight scores were further summarized as summary scores for the three components: Rhe physical component score (PCS), mental component score (MCS), and role component score (RCS) [13].

### 2.3. Evaluation of Nutritional Intake

The brief self-administered dietary history questionnaire (BDHQ) was used to evaluate nutritional intakes. BDHQ collects information on average food intake in the previous one month to assess the habitual intake of nutrients from typical foods [14,15,16]. The questionnaire asks about food intake from 58 types of food, non-alcoholic and alcoholic beverages, daily intakes of rice and miso soup, normal fish and meat recipes, and general diet. Based on BDHQ, the intake of energy, protein, carbohydrate, lipid, and 14 vitamins: Retinol, β-carotene equivalent, retinol equivalent, vitamin D, α-tocopherol, vitamin K, vitamin B1, vitamin B2, niacin, vitamin B6, vitamin B12, folic acid, pantothenic acid, and vitamin C, was calculated. The intake was estimated using a computer algorithm for the BDHQ, which is based on the Standard Tables of Food Composition in Japan 2010 [14,15,16]. The use of nutritional supplements may lead to an excessive intake of micronutrients, such as minerals or vitamins. This may affect to the interpretation of the results. To avoid such a bias, mineral or vitamin intake from supplements was not included in the calculation. Participants who reported a total energy intake of less than 600 kcal/day (half of the energy intake required in the low physical activity category) or 4000 kcal/day or more (1.5-fold the energy intake required in the moderate physical activity category) were excluded from the analysis because of potential biases in the analysis [17,18].

### 2.4. Questionnaire on the Demographics

Participants completed a self-administered questionnaire on their socioeconomic status, lifestyle, and medical history. It included question about age, sex, working/volunteer, living condition, education, exercise/hobby, smoking, alcohol intake, weight/height, and medical treatment of diabetes, hyperlipidemia, or hypertension. The height and weight were used to calculate body mass index (BMI) as: weight (kg)/height (m)^2^. Based on the World Health Organization reference [19], we defined overweight/obese as BMI of 25 or more. We dichotomized variables of working/volunteer, living condition, exercise/hobby, smoking, alcohol intake, and medical treatment into whether or not doing working/volunteer, living alone, doing exercise/hobby, currently smoking, taking alcohol, and having medical treatment.

### 2.5. Statistical Analysis

Means and standard deviations were used to describe the characteristics of participants. The t-test was performed to compare characteristics between men and women. An analysis of covariance (ANCOVA) was conducted to compare vitamin intake between low and high score groups in SF-36. ANCOVA was adjusted by age, occupation/volunteer, living status, education, exercise/hobbies, smoking, drinking, BMI, diabetes treatment, hyperlipidemia treatment, and hypertension treatment. A multiple regression analysis was performed to investigate the relationship between SF-36 scores and vitamin intake. To examine the possibility of the relationship between vitamins intake and SF-36 scores, the significance of differences was defined as *p* < 0.05. SF-36 summary scores were used as outcome variables, whereas vitamin intake was used as an explanatory variable, with adjustments for age, occupation/volunteer, living status, education, exercise/hobby, smoking, drinking, BMI, diabetes treatment, hyperlipidemia treatment, and hypertension treatment. The IBM Statistical Package for Social Sciences (SPSS) Version 24.0 (IBM, Armonk, NY, USA) was used for the analysis. To confirm the relationship between vitamins intake and SF-36 scores, the significance of differences was defined as *p* < 0.003 (0.05/14), preforming the Bonferroni correction based on hypothesis of 14 vitamins.

## 3. Results

### 3.1. Particpant Characteristics

The characteristics of participants are shown by sex (Table 1). Significant sex differences were observed in age, employment status, living status, years of education, exercise/hobby, smoking, alcohol intake, overweight/obesity, diabetes treatment, hyperlipidemia treatment, and hypertension treatment between men and women. PCS and RCS scores were significantly higher in men than in women. No sex differences were observed in MCS scores. Furthermore, a significant sex difference was noted in nutritional intake, with the exception of vitamin B12. The mean food intake and the food sources of the vitamins are shown in Appendix A.

### 3.2. Differences in Nutritional Intake between High and Low PCS/RCS/MCS Groups by Sex

Vitamin intake did not significantly differ between men with lower and higher PCS scores. Retinol, vitamin K, vitamin B2, niacin, vitamin B6, vitamin B12, folic acid, pantothenic acid, and vitamin C intakes showed significant differences between women with lower and higher PCS scores (Table 2). These results imply the intake of significantly lower levels of these nutrients in the lower PCS group than in the higher PCS group. Among men, vitamin B2 intake was significantly lower in the lower MCS group than in the higher MCS group. Among women, the intake of niacin and vitamin B6 was significantly lower in the low MCS group than in the higher MCS group (Table 3). None of the nutrients examined showed a significant difference between men or women with lower and higher RCS scores (Table 4).

### 3.3. Multiple Regression Analysis

A multiple regression analysis of individual vitamin intake and the summary scores showed no correlation with PCS scores in men, whereas a positive correlation was observed between all vitamin intake and PCS scores in women (Table 5). In men, no correlations were found between vitamin intakes and MCS scores. In contrast, in women, positive correlations were observed between vitamin B6, folic acid, vitamin C, and MCS scores (Table 6). Female PCS and MCS scores were strongly associated with vitamin intake.

## 4. Discussion

We herein investigated the relationship between vitamin intake and the summary score of SF-36 in 3162 residents of Shika town. The results obtained revealed that the intakes of retinol, vitamin K, vitamin B2, niacin, vitamin B6, vitamin B12, folic acid, pantothenic acid, and vitamin C were significantly lower in the female low PCS group than in the high PCS group. Vitamin B2 intake was significantly lower in the male lower MCS group than in the higher MCS group. The intake of niacin and vitamin B6 was significantly lower in the female low MCS group than in the higher MCS group. A multiple regression analysis of vitamin intake and summary scores showed no correlation with PCS scores in men. On the other hand, a positive correlation was observed between all vitamin intakes and PCS scores in women. In men, no correlations were found between vitamin intakes and MCS scores. On the other hand, positive correlations were observed between vitamin B6, folic acid, vitamin C, and MCS scores in women. Female PCS and MCS scores were strongly associated with vitamin intake. MCS scores were strongly associated with vitamin B6 and niacin intakes.

Depression is one of the most common diseases worldwide, affecting an estimated 322 million individuals [20], with an increase of 18.4% between 2005 and 2015 [21]. Depression in middle-aged and elderly individuals is often associated with emotional instability, and significantly reduces QOL [22,23,24]. Additionally, the prevalence of depression is generally higher in women. The prevalence of depression in middle-aged and elderly individuals in South Korea was previously reported to be 4.1% in men and 11.7% in women [25]. A study on middle-aged and elderly individuals in China revealed that women were 1.98-fold more likely to have depression than men [26]. In a Japanese study, 4.3% of men and 6.3% of women had severe depressive symptoms [27]. 

Proper eating habits have been identified as an important factor influencing physical and mental health and are closely associated with QOL [3]. Furthermore, irregular eating habits not only interfered with physical health, but also affected the psychological state of individuals and caused emotional instability. These findings suggest that diet quality is associated with the risk of developing depression and QOL. A relationship was previously proposed between a regular diet and PCS and MCS in middle-aged and elderly individuals in China, and a proper diet was important for maintaining QOL [4].

A previous study reported that the prevalence of depression was higher among middle-aged and elderly individuals with a low vitamin intake than among those with a high vitamin intake [11]. Mikkelsen et al. demonstrated that a deficiency in B vitamins, such as vitamins B1, B3, B6, B9, and B12, was associated with depression [28,29]. Furthermore, carotene and vitamin C intakes were associated with the amelioration of depressive symptoms in middle-aged and elderly individuals in Japan [30]. Therefore, diet, particularly the intake of vitamins, plays a major role in health. Vitamin deficiency may lead to poor QOL due to mental illness, depression, altered memory function, and cognitive decline. However, this has not yet been sufficiently demonstrated epidemiologically [20,31]. Furthermore, although a causal relationship has not yet been elucidated, a relationship has been suggested to exist between depression and overweight/obesity. One of the possible etiologies for this relationship is a dietary imbalance. Thao et al. [11] showed that among older Japanese individuals, particularly overweight women, those with depressive symptoms had lower vitamin intakes than those without. The intake of six out of seven B vita-mins (excluding vitamin B3), vitamin C, vitamin K, retinol, α-tocopherol, and cryptoxanthin has been associated with depressive symptoms [11]. In the present study, PCS and MCS scores in women were strongly associated with vitamin intake. Gougeon et al. finding indicated that a higher dietary intake of vitamin B6 among women and vitamin B12 among men is protective against late-life depression among generally healthy seniors living in the community [32]. An insufficient vitamin intake may have physical and psychological effects. Obese individuals with chronic pain are at an increased risk of developing depression. Furthermore, vitamin deficiency may cause depression, which reduces QOL. Therefore, depression and chronic pain are closely related to vitamin intake. The possible mechanisms underlying the impact of overall diet quality on mental health and QOL are explained via oxidative stress and inflammation. An antioxidant-rich diet can act to slow or prevent aging-related pathophysiological and cognitive changes [3].

However, the relationship between vitamin intake and QOL, with a focus on sex differences, remains unclear. To the best of our knowledge, this is the first study to report a relationship between vitamin intake and QOL with sex differences utilizing a large dataset.

This study has several limitations. First, the self-administered dietary history questionnaire is not a very accurate technique for determining nutrients intake. For instance, obese people have possibility to minimize their intake and thin people to over declare quantities of food. In addition, women might always be on a diet and report fewer intakes than the real ones. On the other hand, people in the survey may declare a healthier diet than the real one. The validation study on the BDHQ [16] showed it had ability to estimate mean values for only a limited number of nutrients. However, the study proved it had satisfactory ranking ability for the energy-adjusted intakes of many nutrients among the Japanese population. Second, being a cross-sectional study design, we could not establish a causal relationship. It is not possible to know if depression is aggravated by low vitamin intake or if they ingest few vitamins because of the depression.

## 5. Conclusions

In women, the analysis showed a positive correlation between all vitamin intake and PCS scores, and the analysis of vitamin intake and MCS scores revealed a positive correlation between vitamin B6, folic acid, vitamin C, and MCS scores. In consideration of depression as MCS and chronic pain as PCS, an insufficient vitamin intake may affect QOL in women; however, a causal relationship has not yet been demonstrated. 

## Figures and Tables

**Table 1 nutrients-13-01023-t001:** Characteristics.

	Male (*n* = 1429)	Female (*n* = 1733)	*p*-Value *^4^
	Mean (*n*) ± SD (%)	Mean (*n*) ± SD (%)	
Age	63.27 ± 12.24	64.54 ± 13.11	0.005
Without working/volunteer	802 (57.33%)	1057 (62.18%)	0.006
Living alone	105 (7.41%)	178 (10.41%)	0.003
Education (year)	11.84 ± 3.00	11.42 ± 2.55	0.000
Without exercise/hobby	832 (59.86%)	1113 (66.73%)	0.000
Smoking	460 (32.53%)	110 (6.46%)	0.000
Alcohol intake	1090 (80.5%)	847 (56.39%)	0.000
Overweight/obesity	415 (29.04%)	349 (20.14%)	0.000
Diabetes treatment	146 (10.22%)	106 (6.12%)	0.000
Hyperlipidemia treatment	132 (9.24%)	242 (13.96%)	0.000
Hypertension treatment	428 (29.95%)	431 (24.87%)	0.001
PCS *^1^	45.97 ± 12.41	42.92 ± 14.78	0.000
MCS *^2^	51.3 ± 9.72	51.77 ± 9.91	NS
RCS *^3^	49.8 ± 12.27	48.27 ± 13.14	0.001
Energy (kcal/day)	2013.84 ± 617.89	1641 ± 517.08	0.000
Protein (% energy)	14.60 ± 3.19	15.69 ± 3.12	0.000
Carbohydrate (% energy)	53.43 ± 9.31	55.68 ± 8.17	0.000
Lipid (% energy)	23.41 ± 5.91	26.03 ± 5.87	0.000
Retinol (μg/1000 kcal)	212.1 ± 326.66	182.48 ± 124.39	0.001
β carotene (μg/1000 kcal)	1521.44 ± 1070.95	2108.19 ± 1427.03	0.000
Retinol equivalent (μg/1000 kcal)	340.7 ± 341.88	360.01 ± 182.18	NS
Vitamin D (μg/1000 kcal)	8.15 ± 5.16	8.77 ± 5.17	0.001
α tocopherol (mg/1000 kcal)	3.48 ± 0.99	4.04 ± 1.10	0.000
Vitamin K (μg/1000 kcal)	141.28 ± 80.57	170.49 ± 100.35	0.000
Vitamin B1 (mg/1000 kcal)	0.37 ± 0.09	0.43 ± 0.10	0.000
Vitamin B2 (mg/1000 kcal)	0.63 ± 0.20	0.7 ± 0.20	0.000
Niacin (mg/1000 kcal)	9.07 ± 2.74	9.55 ± 2.73	0.000
Vitamin B6 (mg/1000 kcal)	0.65 ± 0.18	0.71 ± 0.19	0.000
Vitamin B12 (μg/1000 kcal)	5.71 ± 3.16	5.86 ± 2.97	NS
Folic acid (μg/1000 kcal)	156.53 ± 63.21	179.96 ± 72.82	0.000
Pantothenic acid (mg/1000 kcal)	3.2 ± 0.72	3.55 ± 0.75	0.000
Vitamin C (mg/1000 kcal)	48.23 ± 26.73	62.81 ± 31.90	0.000

*1 Physical Component Summary; *2 Mental Component Summary; *3 Role Component Summary; *4 *t*-tests for continuous variables chi-square tests for categorical variables between the sexes. NS: *p* ≥ 0.05.

**Table 2 nutrients-13-01023-t002:** Comparisons of the nutrient intakes according to Physical Component Summary (PCS) groups.

	Male			Female		
	Lower Score Group (*n* = 679)	Higher Score Group (*n* = 546)	*p*-Value *	Lower Score Group (*n* = 807)	Higher Score Group (*n* = 524)	*p*-Value *
	Mean ± SD	Mean ± SD		Mean ± SD	Mean ± SD	
Retinol (μg/1000 kcal)	203.74 ± 205.7	225.98 ± 465.23	NS	177.52 ± 107.46	189.52 ± 137.59	0.003
β-carotene (μg/1000 kcal)	1580.84 ± 1030.95	1418.43 ± 1028.89	NS	2074.99 ± 1357.10	2072.09 ± 1427.63	NS
Retinol equivalent (μg/1000 kcal)	337.26 ± 224.24	345.98 ± 476.6	NS	352.28 ± 169.28	364.14 ± 189.46	0.005
Vitamin D (μg/1000 kcal)	8.23 ± 5.02	7.93 ± 5.13	NS	8.70 ± 4.89	8.63 ± 5.41	NS
α-tocopherol (mg/1000 kcal)	3.51 ± 1.01	3.42 ± 0.95	NS	4.01 ± 1.05	4.08 ± 1.09	NS
Vitamin K (μg/1000 kcal)	145.59 ± 83.69	136.25 ± 79.45	NS	165.87 ± 97.92	175.22 ± 100.72	0.018
Vitamin B1 (mg/1000 kcal)	0.37 ± 0.09	0.36 ± 0.09	NS	0.42 ± 0.10	0.43 ± 0.10	NS
Vitamin B2 (mg/1000 kcal)	0.64 ± 0.19	0.62 ± 0.21	NS	0.69 ± 0.20	0.72 ± 0.19	0.007
Niacin (mg/1000 kcal)	8.98 ± 2.71	9.19 ± 2.67	NS	9.36 ± 2.69	10.00 ± 2.77	0.016
Vitamin B6 (mg/1000 kcal)	0.66 ± 0.18	0.64 ± 0.17	NS	0.7 ± 0.19	0.72 ± 0.20	0.023
Vitamin B12 (μg/1000 kcal)	5.77 ± 3.13	5.63 ± 3.14	NS	5.82 ± 2.84	5.91 ± 3.12	0.036
Folic acid (μg/1000 kcal)	159.44 ± 59.3	152.29 ± 67.31	NS	176 ± 70.56	182.08 ± 71.93	0.006
Pantothenic acid (mg/1000 kcal)	3.21 ± 0.71	3.15 ± 0.74	NS	3.53 ± 0.75	3.59 ± 0.74	0.008
Vitamin C (mg/1000 kcal)	49.53 ± 25.66	45.23 ± 25.89	NS	61.22 ± 30.51	61.9 ± 32.57	0.011

* Analysis of covariances (ANCOVA) adjusted by age, working/volunteer, living alone, education, exercise/hobby, smoking, alcohol intake, overweight/obesity, diabetes treatment, hyperlipidemia treatment, and hypertension treatment. NS: *p* ≥ 0.05.

**Table 3 nutrients-13-01023-t003:** Comparisons of the nutrient intakes according to Mental Component Summary (MCS) groups.

	Male			Female		
	Lower Score Group (*n* = 679)	Higher Score Group (*n* = 546)	*p*-Value *	Lower Score Group (*n* = 807)	Higher Score Group (*n* = 524)	*p*-Value *
	Mean ± SD	Mean ± SD		Mean ± SD	Mean ± SD	
Retinol (μg/1000 kcal)	197.80 ± 207.59	226.52 ± 426.76	NS	181.54 ± 121.07	182.79 ± 119.81	NS
β-carotene (μg/1000 kcal)	1407.09 ± 921.17	1590.77 ± 1109.10	NS	1973.83 ± 1379.04	2150.87 ± 1385.15	NS
Retinol equivalent (μg/1000 kcal)	316.86 ± 223.56	360.87 ± 438.75	NS	347.86 ± 180.59	363.95 ± 174.92	NS
Vitamin D (μg/1000 kcal)	7.89 ± 4.76	8.25 ± 5.31	NS	8.24 ± 4.59	9.01 ± 5.43	NS
α-tocopherol (mg/1000 kcal)	3.40 ± 0.94	3.53 ± 1.02	NS	3.98 ± 1.03	4.08 ± 1.10	NS
Vitamin K (μg/1000 kcal)	131.33 ± 74.03	149.62 ± 87.01	NS	161.77 ± 90.86	175.54 ± 104.67	NS
Vitamin B1 (mg/1000 kcal)	0.36 ± 0.08	0.37 ± 0.09	NS	0.42 ± 0.10	0.43 ± 0.10	NS
Vitamin B2 (mg/1000 kcal)	0.60 ± 0.18	0.65 ± 0.21	0.045	0.69 ± 0.18	0.71 ± 0.21	NS
Niacin (mg/1000 kcal)	8.99 ± 2.63	9.14 ± 2.74	NS	9.45 ± 2.55	9.74 ± 2.87	0.021
Vitamin B6 (mg/1000 kcal)	0.63 ± 0.17	0.66 ± 0.18	NS	0.69 ± 0.18	0.72 ± 0.20	0.026
Vitamin B12 (μg/1000 kcal)	5.58 ± 3.04	5.81 ± 3.21	NS	5.67 ± 2.82	6.00 ± 3.04	NS
Folic acid (μg/1000 kcal)	147.81 ± 52.44	163.12 ± 69.83	NS	171.98 ± 66.31	183.33 ± 74.31	NS
Pantothenic acid (mg/1000 kcal)	3.10 ± 0.66	3.25 ± 0.77	NS	3.49 ± 0.68	3.60 ± 0.79	NS
Vitamin C (mg/1000 kcal)	44.42 ± 22.85	50.21 ± 27.78	NS	58.03 ± 28.31	64.15 ± 33.24	NS

* Analysis of covariances (ANCOVA) adjusted by age, working/volunteer, living alone, education, exercise/hobby, smoking, alcohol intake, overweight/obesity, diabetes treatment, hyperlipidemia treatment, and hypertension treatment. NS: *p* ≥ 0.05.

**Table 4 nutrients-13-01023-t004:** Comparisons of the nutrient intakes according to Role Component Summary (RCS) groups.

	Male			Female		
	Lower Score Group (*n* = 679)	Higher Score Group (*n* = 546)	*p*-Value *	Lower Score Group (*n* = 807)	Higher Score Group (*n* = 524)	*p*-Value *
	Mean ± SD	Mean ± SD		Mean ± SD	Mean ± SD	
Retinol (μg/1000 kcal)	227.75 ± 246.40	205.83 ± 390.80	NS	188.54 ± 140.43	177.86 ± 103.88	NS
β-carotene (μg/1000 kcal)	1573.28 ± 1085.15	1472.50 ± 1001.45	NS	2151.84 ± 1430.59	2019.44 ± 1350.13	NS
Retinol equivalent(μg/1000 kcal)	360.60 ± 263.04	330.36 ± 402.48	NS	369.65 ± 196.31	348.08 ± 162.68	NS
Vitamin D (μg/1000 kcal)	8.63 ± 5.30	7.80 ± 4.92	NS	8.83 ± 5.53	8.57 ± 4.78	NS
α-tocopherol (mg/1000 kcal)	3.52 ± 1.02	3.44 ± 0.97	NS	4.02 ± 1.11	4.04 ± 1.03	NS
Vitamin K (μg/1000 kcal)	147.86 ± 81.48	137.86 ± 82.00	NS	171.61 ± 101.90	168.12 ± 97.14	NS
Vitamin B1 (mg/1000 kcal)	0.37 ± 0.09	0.36 ± 0.09	NS	0.42 ± 0.10	0.43 ± 0.10	NS
Vitamin B2 (mg/1000 kcal)	0.65 ± 0.19	0.62 ± 0.20	NS	0.70 ± 0.20	0.7 ± 0.19	NS
Niacin (mg/1000 kcal)	9.08 ± 2.64	9.07 ± 2.72	NS	9.38 ± 2.86	9.77 ± 2.64	NS
Vitamin B6 (mg/1000 kcal)	0.66 ± 0.18	0.64 ± 0.17	NS	0.71 ± 0.20	0.71 ± 0.19	NS
Vitamin B12 (μg/1000 kcal)	5.98 ± 3.22	5.56 ± 3.08	NS	5.87 ± 3.14	5.84 ± 2.81	NS
Folic acid (μg/1000 kcal)	160.76 ± 61.00	153.76 ± 64.09	NS	180.70 ± 73.41	176.78 ± 69.51	NS
Pantothenic acid (mg/1000 kcal)	3.26 ± 0.72	3.14 ± 0.73	NS	3.55 ± 0.75	3.55 ± 0.74	NS
Vitamin C (mg/1000 kcal)	50.25 ± 28.43	46.15 ± 24.18	NS	63.49 ± 32.71	60.09 ± 30.27	NS

* Analysis of covariances (ANCOVA) adjusted by age, working/volunteer, living alone, education, exercise/hobby, smoking, alcohol intake, overweight/obesity, diabetes treatment, hyperlipidemia treatment, and hypertension treatment. NS: *p* ≥ 0.05.

**Table 5 nutrients-13-01023-t005:** Association between nutrient intakes and Physical Component Summary (PCS).

	Male				Female			
	Standardized Coefficient	95% CI		*p*-Value *	Standardized Coefficient		95% CI	*p*-Value *
	β	Lower	Upper		β	Lower	Upper	
Retinol (μg/1000 kcal)	0.036	0.000	0.003	NS	0.059	0.002	0.012	0.010
β-carotene (μg/1000 kcal)	0.040	0.000	0.001	NS	0.081	0.000	0.001	0.000
Retinol equivalent (μg/1000 kcal)	0.044	0.000	0.003	NS	0.093	0.004	0.011	0.000
Vitamin D (μg/1000 kcal)	0.022	−0.068	0.173	NS	0.075	0.083	0.333	0.001
α-tocopherol (mg/1000 kcal)	0.008	−0.529	0.717	NS	0.077	0.422	1.620	0.001
Vitamin K (μg/1000 kcal)	0.048	0.000	0.015	NS	0.108	0.009	0.022	0.000
Vitamin B1(mg/1000 kcal)	−0.007	−7.937	5.993	NS	0.082	5.162	18.057	0.000
Vitamin B2(mg/1000 kcal)	0.025	−1.630	4.659	NS	0.119	5.282	11.749	0.000
Niacin (mg/1000 kcal)	0.027	−0.100	0.345	NS	0.084	0.199	0.664	0.000
Vitamin B6 (mg/1000 kcal)	0.032	−1.356	5.714	NS	0.098	3.839	10.492	0.000
Vitamin B12 (μg/1000 kcal)	0.010	−0.156	0.231	NS	0.078	0.161	0.588	0.001
Folic acid (μg/1000 kcal)	0.061	0.002	0.022	NS	0.127	0.016	0.034	0.000
Pantothenic acid (mg/1000 kcal)	0.031	−0.337	1.386	NS	0.108	1.190	2.914	0.000
Vitamin C (mg/1000 kcal)	0.053	0.000	0.049	NS	0.120	0.033	0.075	0.000

* Multiple regression analysis adjusted by age, working/volunteer, living alone, education, exercise/hobby, smoking, alcohol intake, overweight/obesity, diabetes treatment, hyperlipidemia treatment, and hypertension treatment. NS: *p* ≥ 0.003.

**Table 6 nutrients-13-01023-t006:** Association between nutrient intakes and Mental Component Summary (MCS).

	Male				Female			
	Standardized Coefficient	95% CI		*p*-Value *	Standardized Coefficient	95% CI		*p*-Value *
	β	Lower	Upper		β	Lower	Upper	
Retinol (μg/1000 kcal)	0.012	−0.001	0.002	NS	0.026	−0.002	0.006	NS
β-carotene (μg/1000 kcal)	0.061	0.000	0.001	NS	0.064	0.000	0.001	NS
Retinol equivalent (μg/1000 kcal)	0.025	−0.001	0.002	NS	0.059	0.000	0.006	NS
Vitamin D (μg/1000 kcal)	−0.006	−0.117	0.094	NS	0.066	0.027	0.225	NS
α-tocopherol (mg/1000 kcal)	0.036	−0.189	0.904	NS	0.064	0.109	1.057	NS
Vitamin K (μg/1000 kcal)	0.076	0.002	0.016	NS	0.075	0.002	0.012	NS
Vitamin B1 (mg/1000 kcal)	0.029	−3.021	9.199	NS	0.065	1.214	11.435	NS
Vitamin B2 (mg/1000 kcal)	0.045	−0.530	4.985	NS	0.054	0.096	5.252	NS
Niacin (mg/1000 kcal)	0.035	−0.070	0.320	NS	0.071	0.067	0.435	NS
Vitamin B6 (mg/1000 kcal)	0.033	−1.258	4.946	NS	0.085	1.607	6.882	0.002
Vitamin B12 (μg/1000 kcal)	−0.009	−0.197	0.143	NS	0.051	−0.003	0.336	NS
Folic acid (μg/1000 kcal)	0.068	0.002	0.019	NS	0.081	0.004	0.018	0.003
Pantothenic acid (mg/1000 kcal)	0.050	−0.089	1.422	NS	0.068	0.201	1.572	NS
Vitamin C (mg/1000 kcal)	0.042	−0.006	0.038	NS	0.087	0.010	0.044	0.001

* Multiple regression analysis adjusted by age, working/volunteer, living alone, education, exercise/hobby, smoking, alcohol intake, overweight/obesity, diabetes treatment, hyperlipidemia treatment, and hypertension treatment. NS: *p* ≥ 0.003.

## Data Availability

The data presented in this study are available on request from the corresponding author. The data are not publicly available due to privacy and ethical policy.

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
