# Peer review of "Relationship between Vitamin Intake and Health-Related Quality of Life in a Japanese Population: A Cross-Sectional Analysis of the Shika Study"

_nutrients, 2021, doi:10.3390/nu13031023_

Round 1

Reviewer 1 Report

This paper provides the description data of various vitamins intake among Japanese residents in 4 select districts. The cross-sectional design may limit the implication, however, it still provide has some importance in filling the research gap related to vitamins intake and quality of life. Some major comments are listed below;

1, how representative is the study population? Are these 4 districts selected by random? 

2, in part 3.3, reading the information from the Table 5 & 6, it should be linear regression, rather than logistic regression. The outcomes here were scores (continuous variables). 

This paper is carelessly prepared, minor comments; 

1, please proof the spelling, many words were inserted with "-" in the middle improperly. 

2, correct the subtitle in line 125 "Statistical AnalysisQuestionnaire on the socioeconomic status" to "Statistical Analysis"

3, in table 1, is there any missing in each variable? Apparently, the N (%) of alcohol intake among male is wrong. 8 percent ? 

4, in "5. Conclusions" part, the last sentence is out of place (line 264-266)

Reviewer 2 Report

The authors took up an interesting topic about the relationship between vitamin intake and quality of life. The main advantage of the study is the size of the study group (n=3162). On the other hand, this topic does not appear to be so new.

Reading the work, various careless notes (typos 'exersize'?, punctuation) as if the work was not checked before submission.

Abstract

The aim of the study?
The results should be supported by numerical values and p value

Introduction

The objective is not formulated.

Material and methods

In what age range were the study groups?

BDHQ – how many days were evaluated?

2.5

Can the description be shorter:

"ANCOVA of characteristics was adjusted by sex, age, occupation/volunteer, living status, education, exercise/hobbies, smoking, drinking, BMI, diabetes treatment, hyperlipidemia treatment, and hypertension treatment. ANCOVA of vitamin intake was adjusted by age, occupation/volunteer, living status, education, exercise/hobbies, tuxedo, drinking, BMI, diabetes treatment, hyper-lipidemia treatment, and hypertension treatment. "

Have subjects been evaluated for taking supplements that affect the results? If no, the bias is possible. If so, in what way?

Results

Tables are large and unreadable (column names 'mean  (N)+/-SD (%)' ? ,  units - information crowding). If p is >=0.05, replace the number with NS; it should be improved

Discussion

The Authors should try to explain what are the mechanism responsible for link between some vitamin intake and PCS/MCS

Conclusions

"In women, a multiple logistic regression analysis of vitamin intake and MCS scores revealed a positive correlation between all vitamin intakes, except for retinol and vitamin B12, and MCS scores"

The name of the statistical test shouldn’t be mentioned here

Carelessness

"This section is not mandatory but can be added to the manuscript if the discussion is unusually long or complex. "?

Reviewer 3 Report

In the present study, the authors examined the relationship between test of SF-36 version 2 and vitamin intake in residents aged 40 years and older in Japan.

Table 1 should also show the data on the intake of total calories, proteins, carbohydrates, fats, etc.

The “brief self-administered dietary history questionnaire” is NOT a very accurate technique for determining intake. Obese people tend to minimize their intake and thin people to over declare quantities of food. In addition, women tend to always be on a diet and report fewer intakes than the real ones. In Table 1 it can be observed that the results in women are almost always different from those of men. In Table 2 we observe that the comparisons of the nutrient intakes according to physical component score groups are only significant in women.

On the other hand, people in the survey usually declare a healthier diet than the real one for fear of appearing neglected. In the case of patients with diseases such as diabetes, in which dietary treatment is essential, they usually recite the prescribed diet and not the real one. All of these are major limitations of the study.

The calculation of the amount of vitamins ingested from the questionnaire data is not clear how they did it or which food composition tables they have used.

Being a cross study, it is not possible to know if depression is aggravated by low vitamin intake or if they ingest few vitamins because they are depressed.

You find that niacin and vitamin B6 levels were significantly lower in the lower MCS group than in the high MCS group. I think this result is random after having made multiple comparisons to see what comes out. The method must be the other way around: they first present a hypothesis as to which particular vitamins may be related to a particular part of the quality of life test, and then they test the hypothesis.

Type I error increases with increasing number of comparisons and we can find a falsely significant difference just by chance. For all this, the Bonferroni correction should be performed.

Round 2

Reviewer 3 Report

No further comments.